# Correlation of the Subjective Hip Value with Validated Patient-Reported Outcome Measurements for the Hip

**DOI:** 10.3390/jcm9072179

**Published:** 2020-07-10

**Authors:** David R. Krueger, Vincent J. Leopold, Joerg H. Schroeder, Carsten Perka, Sebastian Hardt

**Affiliations:** 1Department of Orthopaedics, Herzogin Elisabeth Hospital, 38124 Braunschweig, Germany; d.krueger@heh-bs.de; 2Center for Musculoskeletal Surgery, Charité-Universitätsmedizin Berlin, 10117 Berlin, Germany; vincent-justus.leopold@charite.de (V.J.L.); carsten.perka@charite.de (C.P.); 3Department of Trauma Surgery and Orthopedics, BG Hospital Unfallkrankenhaus Berlin, 12683 Berlin, Germany; joerg.schroeder@ukb.de

**Keywords:** patient-reported outcome measures (PROMs), outcome measures, hip joint

## Abstract

Background: The subjective hip value (SHV) was developed as a patient-reported outcome measurement (PROM) that is easily and quickly performed and interpreted. The SHV is defined as a patient’s subjective hip measurement tool expressed as a percentage of an entirely normal hip joint, which would score 100%. The hypothesis is that results of the subjective hip value correlate with the results of the modified Harris hip score and the International Hip Outcome Tool in patients with hip-related diseases. Methods: 302 patients completed the modified Harris hip score (mHHS), the International Hip Outcome Tool (iHot-33) as well as the SHV. The SHV consist of only one question: “What is the overall percent value of your hip if a completely normal hip represents 100%?”. The patients were divided into five different groups depending on the diagnosis. Pearson correlation was used to evaluate the correlation between the different PROMs and linear regression analysis was used to calculate R^2^. Results: 302 complete datasets were available for evaluation. There was a high correlation between the SHV and the iHOT-33 (*r* = 0.847; *r*^2^ = 0.692, *p* < 0.001) and the mHHS (*r* = 0.832; *r*^2^ = 0.717, *p* < 0.001). The SHV showed a medium (*r* = 0.653) to high (*r* = 0.758) correlation with the mHHS and the iHOT-33 in all diagnosis groups. Conclusion: The SHV offers a useful adjunct to established hip outcome measurements, as it is easily and quickly performed and interpreted. The SHV reflects the view of the patient and is independent of the diagnosis. Further research with prospective studies is needed to test the psychometric properties of the score.

## 1. Introduction

Outcome measurement instruments in medicine resemble an important tool for the evaluation of treatments and procedures. 

Objective measures such as radiographic results, range of motion, and strength were used traditionally to measure the result of a certain therapy or surgery [1,2,3,4,5,6,7].

Most of these outcome measures have been designed for patients with hip arthroplasty or hip fractures [1,2,3,4,5,6,7]. Ceiling effects resemble a relevant limitation in those scores, especially in young and active patients, and the functional ability of the patient is not represented sufficiently [1,2,3,4,5,6,7,8,9,10,11,12]. Subjective measurement elements need to be included to reflect the patient’s ability to return to an active lifestyle [2].

In contrast to the traditional outcome measures, patient-reported outcome measurements (PROMs) are completed by the patient and measure the patient’s perspective on his or her general health or specific functional status [13]. The idea of these subjective measurement tools is to create a valid, responsive and reproducible assessment of a patient’s hip function.

There are several measurement tools available for the evaluation of hip function and symptoms [2,4,14,15,16,17,18,19,20]. Still, in most of the existing literature, traditional scores such as the Harris hip score are used [21,22,23,24,25,26,27,28,29,30,31,32,33,34,35,36,37,38,39].

The Harris hip score was developed as a measurement tool for patients with osteoarthritis of the hip and consists of questions for pain, function, range of motion and deformity [4]. The Harris hip score was modified later using only the pain and function section of the score, thus becoming a patient-reported outcome measurement (PROM) [16]. The Harris hip score resembles one of the most commonly used scores in hip procedures [21]. However, the score also has limitations in the prediction of patient satisfaction [11]. Thus, other PROMs such as the International Hip Outcome Tool (iHOT-33) were developed to address outcomes of hip procedures in young and active patients [2,21].

In a systematic review of patient-reported outcome tools used for hip preservation surgery, the International Hip Outcome Tool scored best and was recommended for further use [21]. Still, patient-reported outcome measurements are often time consuming for the patient to fill out and for the observer to evaluate [40]. The International Hip Outcome Tool consist of 33 questions that have to be answered on a 10 cm visual analogue scale and the result has to be calculated.

A patient-reported outcome measurements that is less time consuming and still accurately reflects the patient’s perspective independently of the diagnosis would be helpful in daily practice. Therefore, the subjective hip value (SHV) was developed based on the subjective shoulder value [41]. For the subjective hip value, the patient was simply asked to give his or her affected hip a value between 0 and 100% compared with an unimpaired normal hip.

The subjective hip value was used earlier to report on results after hip preservation surgery, but has not been evaluated before [42,43,44,45,46,47,48,49,50].

Therefore, the primary aim of this study was to evaluate and compare the subjective hip value to two commonly used PROMs in patients with hip disorders. The secondary goal was to compare the scores in subgroups of patients depending on the diagnosis and a healthy control group.

The hypothesis was that results of the subjective hip value correlate with the results of the modified Harris hip score and the International Hip Outcome Tool in patients with hip-related diseases.

## 2. Materials and Methods 

Patients with a history of hip pain and a control group without a history of hip complaints are routinely evaluated using the hip specific questionnaires modified Harris hip score (mHHS), International Hip Outcome Tool (iHot-33) as well as the subjective hip value (SHV) in our clinic.

The scores were executed by the patients during a consultation in the outpatient clinic for hip pathologies. For this retrospective cohort study, all scores that were performed between January and March 2019 and that met the inclusion criteria (see below) were evaluated.

### 2.1. Inclusion Criteria

Patients aged 18 years and older with a diagnosis of osteoarthritis of the hip, femoroacetabular impingement, hip dysplasia, femoral head necrosis, hip joint infections, synovial hip pathologies and pathologies of the hip-supporting muscles and a healthy control group without a history of hip pain were included.

### 2.2. Exclusion Criteria

Patients under the age of 18, patients with a language barrier, patients who were not able to perform the scores (e.g., neurological diseases) and incomplete datasets were excluded. 

### 2.3. The Patients were Divided into Five Different Groups Depending on the Diagnosis

The patients were divided into five different groups depending on the diagnosis: osteoarthritis of the hip (OA), femoroacetabular impingement (FAI), hip dysplasia (HD), others (O), such as femoral head necrosis, infections, synovial hip pathologies and pathologies of the hip-supporting muscles, and a healthy control group (HC) without a history of hip pain. 

Diagnosis was made based on the complaints of the patient, clinical examination and radiological findings. Anteroposterior radiography and a frog leg view of the hip were used to screen for radiologic signs of a hip pathology.

Osteoarthritis was graded according to the Kallgren and Lawrence score with the a.p.-radiograph showing subchondral sclerosis, subchondral cysts, femoral or acetabular osteophytes or joint space narrowing [51]. Femoroacetabular impingement was defined as α-angle ≥50° (cam impingement) or lateral center edge angle ≥39° (pincer impingement) [52]. Hip dysplasia was defined as lateral center edge angle <22° and acetabular index >14° [52]. Femoral head necrosis was diagnosed on the basis of the a.p.-pelvis and frog leg view together with a magnetic resonance imaging (MRI) of the hip, and it was graded according to the international classification of osteonecrosis of the ARCO (Association Research Circulation Osseous) committee on terminology and classification [53]. Infection of the native hip was diagnosed if the patient had intra-articular effusion of the hip joint in the ultrasound and MRI together with a high leukocyte count (>20.000/mL leukocytes) in the synovial fluid aspiration or microbial growth in the synovial fluid. Periprosthetic joint infection (PJI) was defined according to the European Bone and Joint Infection Society criteria [54]. PJI was present diagnosed when at least one of the following criteria was fulfilled: (I) visible purulence around the prosthesis in a preoperative aspirate or intraoperatively (as determined by the surgeon); (II) the presence of a sinus tract; (III) synovial fluid leukocyte count or differential (>2.000/mL leukocytes or >70% granulocytes); (IV) microbial growth in preoperative joint aspirate, (V) periprosthetic tissue or sonication culture of the removed prosthesis components (>50 colony-forming units (CFU)/mL sonication fluid [55]); or (VI) inflammation in periprosthetic permanent tissue sections seen by histopathological examination, defined as a mean of 23 ≥ granulocytes per 10 high-power fields (corresponding to type II and type III according to the Krenn and Morawietz classification) [56,57].

Synovial hip pathologies included pigmented villonodular synovitis and synovial chondromatosis. Synovial pathologies were anticipated when radiographs showed calcified nodules; MRI showed contrast intake, synovial thickening and effusion, diffuse proliferative synovitis and/or nodules or intra-articular cartilaginous nodules and loose bodies [58,59]. Diagnosis was confirmed or ruled out depending on the histological results after surgery later on. Pathologies of the hip-supporting muscles included degenerative tendinopathies, strains, partial and complete ruptures of the proximal hamstring tendons, gluteus medius and minimus tendons, rectus femoris, tensor fasciae latae and sartorius tendons. The healthy control group consisted of patients that presented in the outpatient clinic at the same time without a history of hip pain or hip-related complaints. The diagnosis was made by two experienced orthopedic surgeons after the patients had performed the scores.

The subjective hip value was designed analogous to the subjective shoulder value [41], which resembles a widely used outcome measure in shoulder surgery. The subjective hip value consists of the following question:

“What is the overall percent value of your hip if a completely normal hip represents 100%?”. If the patient had difficulties with the question, the following alternative question was asked: “A completely normal hip would cost you €1000. How much would you be willing to pay for yours?”.

The Harris hip score (HHS) was initially developed as a measurement tool for patients with osteoarthritis of the hip and was later modified, becoming a patient-reported outcome measurement, known as the modified Harris hip score (mHHS). [4,16]. The modified Harris hip score consists of the items pain (max. 44 points) and function (max. 47 points), creating a maximum 91 points [16]. The result is multiplied by 1.1 to provide a possible maximum of 100 points [16]. The function section includes questions regarding the distance the patient can walk, the ability to put on shoes and socks, the ability to use public transportation, the need to use a walking cane or crutches, limping, the ability to climb stairs and the ability to sit on an ordinary chair for one hour [4].

The International Hip Outcome Tool, like the International Hip Outcome Tool (iHOT-33), was developed to address outcomes of hip procedures in young and active patients and consists of 33 questions divided into the four sections: symptoms and functional limitations (SFL, 16 questions, maximum 49%); sports and recreational activities (SRA, 6 questions, maximum 18%); job-related concerns (JRC, 4 questions, maximum 12%); and social, emotional and lifestyle concerns (SELC, 7 questions, maximum 21%) [2].

The section symptoms and functional limitations includes questions about hip and groin pain, stiffness, walking long distances, pain while sitting, standing for long periods of time, getting up and down off the floor, walking on uneven surfaces, stepping over obstacles, climbing up and down stairs, rising from sitting position, taking long strides, getting out and into the car, grinding and clicking in the hip, putting on socks and overall pain in the hip [2]. The section sports and recreational activities contains questions about concerns of maintaining a desired fitness level, pain after activity, concerns about deterioration of hip pain when doing sports, quality of life change by less participation in sport, changing directions during sport and a decrease in performance level [2]. The section job-related concerns includes questions about moving heavy objects at work, crouching and squatting, concerns that the job will make hip problems worse and difficulties at work due to reduced hip mobility [2]. Finally, the section social, emotional and lifestyle concerns contains questions regarding frustration due to hip problems, sexual activities, distraction caused by hip problems, difficulty releasing stress because of hip problems, discouragement caused by hip problems, difficulties carrying children and the overall time that the patient is aware of the hip disability [2].

The questions are completed on a 10 cm/100-point visual analogue scale for each question, creating a maximum score of 3300 points. The result is calculated in percent of the maximum to generate a maximum score of 100% to allow for better comparability. For this investigation, the validated German translation of the International Hip Outcome Tool was used [60].

The study protocol was approved by the institutional Ethical Committee of Charité-Universitätsmedizin Berlin (Berlin, Germany) and was performed in accordance with the Declaration of Helsinki (approval number EA2/078/20).

### 2.4. Statistics

Normal distribution was tested using the Shapiro–Wilk test. Statistical analysis was performed using the Pearson correlation to evaluate the correlation between the different patient-reported outcome measurements in this population. Linear regression analysis was used to calculate the amount of variance in the International Hip Outcome Tool and its sections and the modified Harris hip score that is predictable by the subjective hip value (R^2^). The level of significance was set at *p* = 0.01. Statistical analysis was performed using the IBM SPSS Statistics software (IBM Corp. Released 2013. IBM SPSS Statistics, Version 22.0. Armonk, NY, USA).

## 3. Results

Between January and March 2019, a total of 302 complete datasets were available for evaluation. The patient population consisted of 164 female and 138 male patients with a mean age of 44 (18–92) years (Table 1). Demographic characteristics of the subgroups are displayed in Table 1.

Overall results of the whole patient population were 64 ± 28.4% for the subjective hip value of 59 ± 28.0% for the International Hip Outcome Tool, and 68 ± 25.2 points for the modified Harris hip score (Figure 1). For the osteoarthritis group, the average subjective hip value was 46 ± 23.7%, the International Hip Outcome Tool was 45 ± 18% and the modified Harris hip score was 55 ± 16.5 points (Figure 1). An average subjective hip value of 64 ± 20.8%, an average International Hip Outcome Tool value of 54 ± 20.4% and a modified Harris hip score of 68 ± 21.9 points were seen in patients with femoroacetabular impingement. For the group with developmental dysplasia of the hip, the average subjective hip value was 59 ± 21.1%, the International Hip Outcome Tool value was 45 ± 19.7% and the modified Harris hip score was 57 ± 17.2 points. Patients with other hip-related diseases scored an average subjective hip value of 45 ± 28.1%, an International Hip Outcome Tool score of 46 ± 25.9% and a modified Harris hip score of 52 ± 28,6 points. The average subjective hip value was 98 ± 4.8%, the International Hip Outcome Tool value was 96 ± 6.0% and the modified Harris hip score was 98 ± 4.7 points in the control group without a history of hip pain.

The results did not show normal distribution, as assessed by the Shapiro–Wilk-Test, *p* < 0.001. The SHV showed a high correlation with the iHOT-33 (*r* = 0.847; *p* < 0.001) and the mHHS (*r* = 0.832; *p* < 0.001) (Table 2; Figure 2). The subitems SFL (*r* = 0.821; *p* < 0.001), SRA (*r* = 0.720; *p* < 0.001) and SELC (*r* = 0.700; *p* < 0.001) showed a high correlation with the SHV. Moderate correlation was noted in the subitem JRC (*r* = 0.590; *p* < 0.001).

For patients with osteoarthritis, the correlation was high between the SHV and the mHHS (*r* = 0.711; *p* < 0.001) and moderate between the SHV and the total iHOT-33 (*r* = 0.653; *p* < 0.001) and the SFL of the iHOT-33 (*r* = 0.633; *p* < 0.001) (Table 3). A low correlation was found with the subitems SRA (*r* = 0.318; *p* = 0.015), JRC (*r* = 0.356; *p* = 0.006) and SELC (*r* = 0.438; *p* = 0.001) of the iHOT-33.

The correlation was high between the SHV and the mHHS (*r* = 0.716; *p* < 0.001) and the total iHOT-33 (*r* = 0.746; *p* < 0.001) in patients with femoroacetabular impingement (Table 3). A moderate correlation was seen with the subitems SFL (*r* = 0.691; *p* < 0.001), SRA (*r* = 0.570; *p* < 0.001), and SELC (*r* = 0.620; *p* < 0.001) of the iHOT-33. Only a low correlation was seen with the JRC (*r* = 0.325; *p* = 0.009) section.

Patients with hip dysplasia showed a high correlation between the SHV and the total iHOT-33 (*r* = 0.753; *p* < 0.001) and the SFL section (*r* = 0.734; *p* < 0.001) (Table 3). The mHHS (*r* = 0.669; *p* < 0.001), SRA (*r* = 0.590; *p* < 0.001), and SELC (*r* = 0.537; *p* < 0.001) showed a moderate and JRC (*r* = 0.344; *p* = 0.010) low correlation.

The correlation was high between the SHV and the mHHS (*r* = 0.758; *p* < 0.001), the total iHOT-33 (*r* = 0.755; *p* < 0.001) and the SRA of the iHOT-33 (*r* = 0.723; *p* < 0.001) in patients with other hip-related complaints (Table 3). A moderate correlation was seen with the subitems SFL (*r* = 0.696; *p* < 0.001), and SELC (*r* = 0.587; *p* < 0.001) of the iHOT-33. Only a low correlation was seen with the JRC (*r* = 0.471; *p* = 0.009) section.

In the healthy control group, a high correlation was seen between the SHV and the total iHOT-33 (*r* = 0.777; *p* < 0.001) and a moderate correlation between the SHV and mHHS (*r* = 0.539; *p* < 0.001), SRA (*r* = 0.506; *p* < 0.001) and SELC (*r* = 0.552; *p* < 0.001) section (Table 3). A low correlation was found with the SFL (*r* = 0.380; *p* = 0.001) and no correlation with the JRC (*r* = 0.160; *p* = 0.188) section. 

## 4. Discussion

The subjective hip value offers a valuable additional patient-reported outcome measurement for the evaluation of hip function. It reflects the perspective of the patient and is easily administered and time-saving for the patient and the investigator. Although the subjective hip value has been used to report on results after hip arthroscopy, cartilage implantation of the hip and muscular reconstructions of the hip abductors, it has never been evaluated before [42,43,44,45,46,47,48,49,50]. We could show a high and statistically significant correlation between the subjective hip value and established hip measures such as the modified Harris hip score and the International Hip Outcome Tool. Linear regression analysis showed that a high percentage of variance in the complete patient population could be predicted by the subjective hip value (*r*^2^) (69% for the mHHS, 71% for the iHOT-33).

We could show that the subjective hip value is independent of the diagnosis, as it shows high variance throughout all subgroups. The subjective hip value showed medium (*r* = 0.653) to high (*r* = 0.758) correlation with the modified Harris hip score and the International Hip Outcome Tool in all groups. The correlation was highest with the modified Harris hip score in patients with “other” hip pathologies and lowest with the iHOT-33 in patients with osteoarthritis of the hip.

In contrast to the other groups, in patients with osteoarthritis, there was no significant correlation between the SHV and the sports and recreational activities (SRA) section of the International Hip Outcome Tool (*r* = 0.318, *p* = 0.015). In all other groups the correlation of the subjective hip value and the sports and recreational activities section was moderate to high. One explanation might be the higher average age in this population (66 years) compared to patients with FAI (41 years), hip dysplasia (28 years) or other pathologies (59 years). In contrast to the other groups tested, sport activities might not have the same relevance for patients with osteoarthritis in this study.

The average results in the healthy control group were 97.4% for the subjective hip value, 97.2 points for the modified Harris hip score and 97.7% for the International Hip Outcome Tool. Correlation of the subjective hip value was higher with the International Hip Outcome Tool (*r* = 0.539) compared to the mHHS (*r* = 0.777). The higher range of the results in the International Hip Outcome resemble a potential explanation of this effect. Even in patients without hip problems, the result might not be 100%. Some questions might not only reflect the actual condition of the hip during the last month, but also general concerns about a deterioration in hip function, such as question 25 “How much concerned are you that your job will make your hip worse?” (average score 96.3%) or question 17 “How concerned are you about your ability to maintain your desired fitness level?” (average score 94.6%). Even patients without known hip-related problems might experience some symptoms that are questioned in the International Hip Outcome Tool, like a clicking or grinding in the hip (question 14, average 90.8%). Only 12 out of 70 individuals in the control group scored the best possible result in the International Hip Outcome Tool (44 with SHV and 47 with mHHS).

The modified Harris hip score showed higher average scores in all groups compared with the subjective hip value and International Hip Outcome Tool results, except for the hip dysplasia group. An overestimation of the results in the modified Harris hip score has been described earlier and can be caused by ceiling effects and other limitations, especially in young and active patients [11,61].

A ceiling effect describes the effect when a high proportion of patients score the best possible result, and it can be caused by test items that are not challenging enough for the individuals tested [12,62]. This effect was reported for the Harris hip score in both patients after hip arthroplasty and after hip preservation surgery [11,12,61]. Questions about using or not using walking canes, the ability to use stairs without a railing and the absence of limping might not reflect all needs of an active patient.

In this study, the average score was also higher in the arthroplasty group. Even in this group, the personal expectations and functional needs might not be completely reflected by the modified Harris hip score. Thus, the modified Harris hip score might simply not be able to detect all clinically relevant properties of the patient [12]. In shoulder surgery, the subjective shoulder value was found with the best distribution of outcomes among several shoulder-specific scores with no signs of a ceiling effect [63]. As the subjective hip value does not test specific items, the patient expresses all issues that are important to them in the result.

The effect of higher average scores in the modified Harris hip score compared to the subjective hip value was seen earlier, with the modified Harris hip score improving faster after hip arthroscopy compared to the International Hip Outcome Tool [44]. Higher scores were also seen after surgical dislocation of the hip, but no information was given if the Harris hip score was used in the modified or original version [42].

### Limitations

One important limitation of this study is that this dataset allows no validation of the subjective hip value. Further research is needed to evaluate the reliability, validity and responsiveness in a larger patient population with longitudinal data. The retrospective design of this study resembles another limitation.

## 5. Conclusions

The subjective hip value offers a useful adjunct to established hip outcome measurements, as it is easily and quickly performed and interpreted. The subjective hip value reflects the view of the patient and is independent of the diagnosis. Further research with prospective studies is needed to test the psychometric properties of the score.

## Figures and Tables

**Figure 1 jcm-09-02179-f001:**
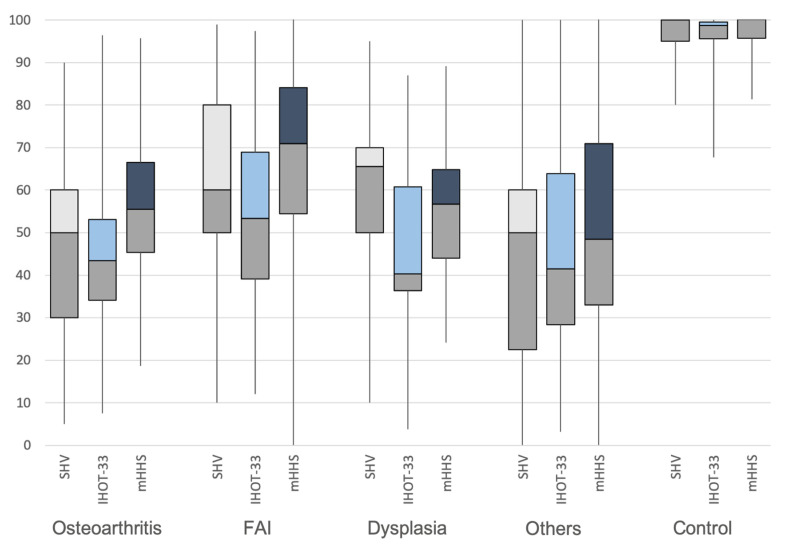
Patient-reported outcome measurement results in the different groups.

**Figure 2 jcm-09-02179-f002:**
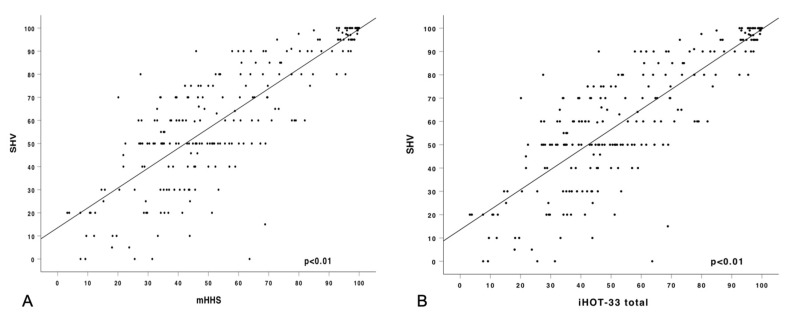
Pearson correlation of the subjective hip value (SHV) with (**A**) modified Harris hip score (mHHS) and (**B**) International Hip Outcome Tool (iHOT-33).

**Table 1 jcm-09-02179-t001:** Patient demographics.

	*N* =	Age (Range) Years	Gender f/m
Osteoarthritis	58	66 (35–87)	27/31
Femoroacetabular impingement (FAI)	65	41 (18–65)	27/38
Dysplasia	56	28 (18–44)	38/18
Others	53	59 (24–92)	33/20
Control	70	30 (20–77)	39/31
Total	302	44 (18–92)	164/138

**Table 2 jcm-09-02179-t002:** Correlation coefficients (*r*) and *r*^2^ of the subjective hip value (SHV) with the modified Harris hip score (mHHS) and the International Hip Outcome Tool (iHOT-33).

	mHHS			iHOT-33 Total		
Group	*r* =	*r*^2^ =	*p* =	*r* =	*r*^2^ =	*p* =
Osteoarthritis	0.711	0.506	0.001	0.653	0.426	0.001
FAI	0.716	0.513	0.001	0.746	0.557	0.001
Dysplasia	0.669	0.448	0.001	0.753	0.567	0.001
Others	0.758	0.575	0.001	0.755	0.570	0.001
Healthy	0.539	0.291	0.001	0.777	0.604	0.001
Total	0.832	0.692	0.001	0.847	0.717	0.001

**Table 3 jcm-09-02179-t003:** Correlation coefficients (*r*) and *r*^2^ of the SHV and the subitems of the International Hip Outcome Tool (iHOT-33). SFL = symptoms and functional limitations, SRA = sports and recreational activities, JRC = job-related concerns, SELC = social, emotional and lifestyle concerns.

	iHOT-33 SFL			iHOT-33 SRA			iHOT-33 JRC			iHOT-33 ELC		
Group	*r* =	*r*^2^ =	*p* =	*r* =	*r*^2^ =	*p* =	*r* =	*r*^2^ =	*p* =	*r* =	*r*^2^ =	*p* =
Osteoarthritis	0.633	0.401	0.001	0.318	0.101	0.015	0.356	0.127	0.006	0.438	0.192	0.001
FAI	0.691	0.477	0.001	0.570	0.325	0.001	0.325	0.106	0.009	0.620	0.384	0.001
Dysplasia	0.734	0.539	0.001	0.590	0.348	0.001	0.344	0.118	0.010	0.537	0.288	0.001
Others	0.696	0.484	0.001	0.723	0.523	0.001	0.471	0.222	0.001	0.587	0.345	0.001
Healthy	0.380	0.144	0.001	0.506	0.256	0.001	0.160	0.026	0.188	0.552	0.305	0.001
Total	0.821	0.674	0.001	0.720	0.518	0.001	0.590	0.348	0.001	0.707	0.500	0.001

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
