# Peer review of "Correlation of the Subjective Hip Value with Validated Patient-Reported Outcome Measurements for the Hip"

_jcm, 2020, doi:10.3390/jcm9072179_

Round 1

Reviewer 1 Report

The aim of this study was to analyse and compare the SHV to commonly used PROMs in patients with different diseases affecting the hip joint. The secondary goal was to compare the score of patient subgroups depending on the diagnosis. Furthermore, the score was compared to a healthy control group.

M&M: The authors do not state any hypothesis in the introduction or M&M section. Maybe this would add some value to the study. There is no clear information about the study design in the abstract or the M&M section. The design of the study was just mentioned once in the limitations section at the end of the manuscript. If the study performed by the authors was of restrospective nature, how did the authors choose the patients that were included in the study? I understand from the text that the study was a retrospective cohort study. What is the standard pre- and postoperative score used in the authors' clinics? Did the authors just choose to test the SHV on 302 randomly selected or consecutive patients and then analyse their results in comparison to standard measurement tools in the field? Please clarify. 

There were five diagnosis groups: OA, HD, FAI, O and HC (healthy control). How did the authors define OA or HD and FAI. Did they use any radiographic criteria (Kellgren & Lawrence Score, or others?, CE-angle, etc.?). How was FAI defined? Please further define synovial hip pathologies and pathologies of the hip supporting muscles. The better the diagnostic groups are defined the better is the discrimination of the study results for the different groups. 

IRB approval number and the IRB agency/institute/University has to be included in the ethical clearance statement. 

Statistics: The statistical methods are sound and well explained. 

Results: The results are well described. Were there any significant differences between diagnostic groups? 

Discussion: The discussion is not deep enough. Please provide a deeper look into the literature. More information about the ceiling effects would be beneficial for interested readers. What is the core finding of the study? What does it contribute to the existing literature? Is it that the SHV is independent of the diagnosis? It has been shown in the past that the SHV is easy to perform and it is known that the SHV reflects the view of the patient.

Author Response

Thank you.

Reviewer 2 Report

This is a prospective study to validate a new hip outcome measure.  

Abstract

- Provide a rationale for this study 

Introduction

- Discussion of the Harris hip score and iHOT can be condensed - Provide a reason for why this hip score was created

Methods

How was the question for the SHV developed? 

- Are iHOT and mHHS validated for the diagnoses you included?

- Please clarify your inclusion and exclusion criteria.  Specifically, who was excluded?

- Who determined the patient's diagnosis and at what point? I'm assuming it was after the SHV and outcome scores were administered.

Results

This section needs to be heavily edited to be more concise. The tables and text are duplicated.

Discussion

Line 218: How does this outcome measure satisfaction? I do not think you can summarize this based on your current study.  Was it compared to another satisfaction outcome?

-What is the clinical relevance of this study? Why did results slightly differ across diagnoses? Are you going to replace your hip-related PROMs with this measure? 

Author Response

Thank you.

Round 2

Reviewer 2 Report

Thank you for making these changes. 

This manuscript is a resubmission of an earlier submission. The following is a list of the peer review reports and author responses from that submission.